# The Effect of Copper Content on the Mechanical and Tribological Properties of Hypo-, Hyper- and Eutectoid Ti-Cu Alloys

**DOI:** 10.3390/ma13153411

**Published:** 2020-08-03

**Authors:** Yiku Xu, Jianli Jiang, Zehui Yang, Qinyang Zhao, Yongnan Chen, Yongqing Zhao

**Affiliations:** 1School of Materials Science and Engineering, Chang’an University, Xi’an 710064, China; 2019131018@chd.edu.cn (J.J.); 2018131004@chd.edu.cn (Z.Y.); zqy_ustb@163.com (Q.Z.); 2Northwest Institute for Nonferrous Metal Research, Xi’an 710016, China; trc@c-nin.com

**Keywords:** Ti-Cu alloys, mechanical properties, tribological properties, microstructure, Ti_2_Cu precipitates

## Abstract

Titanium alloys are widely used in aerospace, chemical, biomedical and other important fields due to outstanding properties. The mechanical behavior of Ti alloys depends on microstructural characteristics and type of alloying elements. The purpose of this study was to investigate the effects of different Cu contents (2.5 wt.%, 7 wt.% and 14 wt.%) on mechanical and frictional properties of titanium alloys. The properties of titanium alloy were characterized by tensile test, electron microscope, X-ray diffraction, differential scanning calorimetry, reciprocating friction and wear test. The results show that the intermediate phase that forms the eutectoid structure with α-Ti was identified as FCC Ti_2_Cu, and no primary *β* phase was formed. With the increase of Cu content, the Ti_2_Cu phase precipitation in the alloy increases. Ti_2_Cu particles with needle structure increase the dislocation pinning effect on grain boundary and improve the strength and hardness of titanium alloy. Thus, Ti-14Cu shows the lowest elongation, the best friction and wear resistance, which is caused by the existence of Ti_2_Cu phases. It has been proved that the mechanical and frictional properties of Ti-Cu alloys can be adjusted by changing the Cu content, so as to better meet its application in the medical field.

## 1. Introduction

Titanium alloys have high strength to weight ratio, excellent burn resistant and outstanding biocompatibility among metallic materials, therefore, they become ideal candidates for aerospace, chemical, biomedical and other important fields [1,2,3,4,5]. The physical and mechanical properties of titanium alloys can be adjusted by adding different alloying elements with various contents to them.

Generally, titanium alloys can be divided into three catalogues: *α, α* + *β* and *β* alloy. By alloying with *β*-stabilizing elements in titanium alloys, it is possible to decrease the liquids temperature and lessen the reactivity of the titanium, thus improve their physical and chemical properties. It is known that copper is a *β*-stabilizing element for titanium alloys. Basically, the addition of Cu to Ti alloys can lower the melting point of the alloy and increase the thermal conductivity [6], which facilitates the enhancement of burn resistance and machinability of titanium alloy. Wang also pointed out that with the increase of copper content, the diffusion of elements in the alloy was accelerated and the melting point of the alloy was reduced [7]. Copper as an eutectoid stable element, can enlarge the phase and make the eutectoid reaction speed very fast, so the phase cannot be stabilized to room temperature. According to the phase diagram of Ti-Cu binary alloy, a Ti-Cu alloy with precipitated Ti_2_Cu phase has a transition temperature of about 1000 °C [8].

The equilibrium phase diagram of the Ti-Cu binary alloys indicates that adding more than 3% copper element into titanium alloy has two phases, forming Ti_2_Cu phase and forming needle structure, which can improve the mechanical properties of the material [9,10]. Depending on the specific copper content added to the binary system, Ti-Cu alloys may show distinct microstructures, e.g., eutectoid, hyper- and hypo-eutectoid microstructures, and this allows the possibility for the mechanical behavior of Ti-Cu alloys to be adjusted by controlling the fraction and distribution of Ti_2_Cu precipitation [11,12]. As a kind of structural material for applications at high temperatures, mechanical properties such as strength and ductility are important for Ti-Cu alloys, and these properties represent high dependence on the addition of copper and processing route [13]. Souza [14] studied the effect of cooling rate on the properties of Ti-Cu alloys, and the results showed that eutectoid grains (*α* + Ti_2_Cu) would be produced at any cooling rate. When the cooling rate is greater than 9 °C·s^−1^, martensite structure will appear, thus increasing the hardness of Ti-Cu alloys. Andrade [15] studied the influence of composition on mechanical properties of as-cast Ti-Cu alloys with copper content ranging from 0.5 to 10 wt.%. It is pointed out that the hardness of Ti-Cu alloys increases with the increase of Cu content because Cu is beneficial to the formation of Ti_2_Cu, the hard metal interphase. In addition, Li [16] pointed out that with the increase of annealing temperature, phase segregation would occur and TiCu_3_ and Ti_3_Cu_4_ intermetallic compounds would appear. This intermetallic compound will make Ti-Cu alloys maintain good mechanical properties.

Studies have shown that adding copper to titanium alloys can promote casting and improve the mechanical properties of products [17,18,19,20]. Considering the addition of copper could significantly modify the composition and microstructure of Ti-Cu alloys, the mechanical behavior and chemical properties may be also varied accordingly. Therefore, the role of Cu in alloy design is of great significance to the development of new alloys. Thus, the purpose of this work is to investigate the effect of copper content and evaluate the role of Ti_2_Cu phase on the mechanical and tribological properties of hypo-, hyper- and eutectoid Ti-Cu alloys.

## 2. Experiment

### 2.1. Materials and Preparation

The Ti-Cu alloys used in this work were comprised of Cu with different concentrations of 2.5, 7.0 and 14.0 wt.%, which were hypo-eutectoid, eutectoid and hyper-eutectoid structure, respectively. The raw materials were supplied by the Northwest Institute of Nonferrous Metals (China). Ti-Cu alloys were first heated in the resistance furnace with the temperature equal to 800–850 °C. Both the heating and holding time were about 30 min. Then the forging was carried out under a 2000 t hydraulic press with 750 kg air hammer. The forging temperature was generally 150-200 °C below the phase transition temperature. The bar with diameter of 40 mm was obtained and then heat treated. Finally, the alloy was quenched in water to improve its hardness. The chemical compositions are listed in Table 1.

### 2.2. Mechanical and Tribological Properties

The mechanical properties of the Ti-Cu alloys were evaluated by tensile testing and Vickers hardness testing. Tensile samples with a diameter of 5 mm and gauge length of 45 mm were machined from the center of forged bars, and wet polished using waterproof emery paper. Tensile tests were performed five times for each alloy at a nominal strain rate of 4.2 × 10^−3^ s^−1^ using a microcomputer controlled electronic universal testing machine-5105 (CMT-5105 ) (Jinan, China) universal testing machine at room temperature. The specimens were etched by Keller’s reagent (10 mL HF + 25 mL HNO_3_ + 15 mL HCl + 500 mL H_2_O). Microstructure examination was conducted under an optical microscope (GX71, Olympus, Japan) and a scanning electron microscopy (SEM, JSM-6700, Tokyo, Japan). Vickers hardness measurements were carried out by a MH-5 Digimatic Vickers hardness tester (Shanghai, China), using a load of 200 g for 30 s. A total of five indents were taken for each sample, and average value of hardness was reported here.

Tribological properties of Ti-Cu alloys (2.5, 7 and 14 wt.%) under ring-on-block contact configuration were evaluated using a reciprocating friction and wear test-rig of MMQ-200. During the test, the coefficient of friction can be continuously and automatically recorded as a function of sliding time. The sliding tests were repeated three times for reliability and reproducibility. The phases were confirmed using a D8 ADVANCE X-ray diffraction (XRD) (Germany) with Cu-kα radiation at a scanning rate of 0.05° per second in the 2θ ranges from 30° to 80° and differential scanning calorimetry (DSC) (EC2000) (USA). After the sliding test, the morphologies and depth profiles of wear tracks were analysis by scanning electron microscopy (SEM) (JSM-6700) (Tokyo, Japan) and transmission electron microscopy (TEM) (FEI TalosF200X TEM) (Hillsboro, OR, USA). Energy dispersive X-ray spectroscopy (EDS) (JSM-6700) (Tokyo, Japan) was selected to determine the elemental distributions. As the friction progressed, the surface of the substrate was ground to a circle with a radius of 5 mm. As the wear time increases, the worn scratch becomes deeper, and the amount of wear also increases. The hardness (HRC) of (GCr 15) bearing steel balls (6 mm diameter) friction couple was 62–63, and the average surface Ra was about 0.01 μm. All the friction and wear tests were performed using a constant normal load of 5 N with a frequency of 5 Hz and an oscillating amplitude of 5 mm for 10 min at room temperature 400 °C, 600 °C and continuous heating at humidity (relative humidity is about 30%).

## 3. Results and discussion

### 3.1. Mechanical Properties of the Ti-Cu Alloys

The ultimate tensile strength (UTS), yield strength (YS), elongation and hardness (HV) of the Ti-Cu alloys are shown in Figure 1.

The true stress–strain curve of Ti-7Cu alloy exhibits a yield point at about 550 MPa and reaches UTS at about 700 MPa, indicating a typical ductility (Figure 1a). For Ti-14Cu alloy, YS and UTS rise to about 600 MPa and 790 MPa, respectively. Compared with Ti-2.5 Cu alloy shown in Figure 1b, YS and UTS of Ti-14Cu alloy increased by 30% and 25%, respectively. This is basically consistent with the former research that results of this study that the yield points of Ti-5Cu and Ti-15Cu are 522–592 MPa [21]. However, opposite to the increased strength, the elongation decreases with the increase of Cu content, as shown in Figure 1c. A relatively low ductility of Ti-14Cu alloy is measured by 12% elongation which is about 3% lower than that of Ti-7Cu and greatly lower than Ti-2.5Cu alloy (30% elongation) [22]. It can be deduced that the addition of Cu not only plays a significant role on the physiochemical property which improves the burn resistance, but also alters the mechanical properties. The elongation is tended to decrease with increasing the copper concentration. The lowest elongation of Ti-Cu alloys is at the alloy having 14% Cu. The hardness test results in Figure 1d shows that the hardness value increases with the increase of Cu content from 2.5 wt.% to 14 wt.%. The hardness of Ti-7Cu and Ti-14Cu increased by 22% and 43%, respectively compared with Ti-2.5Cu.

Figure 2 shows the fracture morphology of the Ti-Cu alloys. As can be seen, the Ti-2.5Cu alloy has dimple fracture and has ductile fracture characteristics, which is a typical plastic fracture.

The Ti-7Cu alloy has obvious step and dimple structure, which is ductile and brittle mixed fracture. The Ti-14Cu alloy is dominated by step structure and is a typical cleavage fracture. The reduction of area increases with the increase of copper content. Such a low elongation rate is traded with the high strength which could bring challenges to the formability and creep resistance of the alloy. As a result, typical intergranular fracture and demulsification grains can be clearly observed by observing the fracture morphology of titanium alloy with different copper content (Figure 2a–c) and the corresponding fracture cross-section morphology (Figure 2d–f), respectively.

In order to further explore the influence of section microstructure on the properties of Ti-Cu alloys, the grain size of section microstructure was measured by Image Pro Plus 6.0 (1993, 2003, Media Cybernetics, MD, USA). Each group was measured 5 times, and the average value was taken as the final grain size. The results are shown in Table 2.

The higher the Cu content is, the larger the grain size of the Ti-Cu alloys would be. Based on SEM characterization of the microstructure in Figure 2, it can be concluded that the grain boundary of Ti-2.5Cu alloy is not obvious, with the smallest grain size of about 1.198 µm. The grain boundary of Ti-7Cu alloy is obvious, and the grain size is about 10.582 µm. Displaying typical large-grain boundary structure, the grain size of Ti-14Cu alloy increases significantly to 56.934 µm, which is 81.41% and 97.90%, respectively compared with Ti-7Cu alloy and Ti-2.5Cu alloy. It is deduced that the higher Cu content may bring higher diffusion, and give rise to larger eutectoid lamellar space and grain size. The results are consistent with Ref. [21]. It can be concluded that the fracture is governed by cleavage mechanism, with the appearance of crack propagation due to the rupture of atomic bonds close to some crystallographic planes. Then intergranular fractures occur when length of cracks propagating along the grain boundaries reach a critical limit.

### 3.2. Tribological Properties of the Ti-Cu Alloys

Figure 3 shows the XRD patterns and DSC heating curves of titanium alloys with different Cu contents.

It is proved that the phases of Ti-2.5Cu, Ti-7Cu and Ti-14Cu are mainly *α*-Ti and Ti_2_Cu, and the precipitation amount of Ti_2_Cu increases with the increase of Cu content. This indicates that Cu content increases the volume fraction of eutectoid structure, thus increasing the content of Ti_2_Cu intermetallic phase. The phase transition trend is consistent with the DSC results in Figure 3b. Due to melting heat absorption, titanium alloys appear as pits at around 690 °C. The eutectoid transformation of the Ti-Cu binary alloy from *β*-Ti to *α*-Ti and Ti_2_Cu can be found at 800 °C. The peritectic reaction is proved to be occurring at about 1000 °C. The results are also consistent with the phase transition process of Ti-Cu binary alloy phase diagram. Based on the above, it can be concluded that the addition of Cu in titanium alloys effectively promoted the eutectoid transformation process, thus increasing the precipitation amount of Ti_2_Cu phase.

The friction coefficients of Ti-Cu alloys at room temperature, 400 °C and 600 °C with continuous heating condition are exhibited in Figure 4a–c, respectively.

The friction coefficients curves of different Ti-Cu alloys exhibit similar friction behavior at the same temperature which shows a trend of increasing at first and then stabilizing later. The Ti-7Cu alloy exhibits a relatively high friction coefficient with a relatively stable value of 0.315, while the friction coefficient of the Ti-14Cu is 68%, which is lower than that of Ti-7Cu at room temperature. After a pre-grinding period, the surface is gradually worn flat and the friction coefficient between the grinding surfaces increases and finally reaches the steady state. Copper added to titanium alloy will be preferentially precipitated at the grain boundary to form the copper-rich region, and then form Ti_2_Cu with a BCC structure. As a high hardness intermetallic compound, Ti_2_Cu increases the hardness of alloys. Under the condition of friction, the high load of rigid Ti_2_Cu has a certain bearing capacity, which improves the wear resistance of the materials. As the temperature reaches 400 °C and 600 °C, the friction coefficient of the Ti-14Cu fluctuates around a relative lower level than that of Ti-2.5Cu and Ti-7Cu alloys. This may be caused by the alloying of copper in titanium alloys. Copper alloying inhibits grain coarsening in terms of kinetics and thermodynamics so that the Ti-14Cu alloy has higher thermal stability and thus reduces the amplitude of the friction coefficient during friction.

Figure 4d shows a schematic diagram of reciprocating friction and wear test. Based on above, the Ti-14Cu alloy exhibits superior friction reducing than Ti-2.5Cu and Ti-7Cu alloys. It is probably due to the reason that Cu is a kind of good lubricant for titanium alloys. Adding copper into Ti can effectively form a hard Ti_2_Cu phase in the friction process. The hard phase (Ti_2_Cu) is mainly involved in the abrasive wear during the friction process under the condition of friction because the combination of Ti_2_Cu phase and matrix is very weak and easy to peel off. Further, the furrow is easily generated by friction between friction pairs, while the titanium alloy is easy to produce adhesive wear due to low surface hardness and insufficient plastic shear resistance. When the copper content is low, the wear process is mainly adhesive wear due to low Ti_2_Cu content and the wear degree is relatively large. The wear resistance of titanium alloy can be improved by producing hard metal intermetallic phase in titanium alloy and introducing abrasive wear to offset part of adhesive wear.

The average friction coefficients of Ti-Cu alloys are present in Figure 5a.

It can be found that the average friction coefficient increases as the temperature increase for the Ti-Cu alloys with the same Cu content. The change of mechanical and physical properties is attributed to the increase of intermolecular distance caused by thermal expansion, which leads to the increase of viscosity and the adhesion friction coefficient of titanium alloys. It has also been reported that high temperatures promote the diffusion of oxygen atoms, leading to the oxidation of Ti and Cu atoms in the metal [23]. TiO_2_ oxide is also a kind of hard phase which may aggravate abrasive wear. It may also be a reason that the Ti-14Cu alloy produces less titanium oxide within the same frictional area reducing the abrasive wear process.

In addition, the friction properties of the three alloys under the condition of continuous temperature rise were also recorded in Figure 5. It can be seen that at room temperature, the friction coefficients of Ti-2.5Cu and Ti-7Cu alloys are very high and maintained a relatively similar trend showing a relatively large fluctuation. When the temperature reaches to 400 °C and 600 °C, the friction coefficient is observed to decrease and the value of the Ti-2.5Cu alloy is much lower than the Ti-7Cu alloy. As is known, it is easy to form titanium oxide which is a hard particle but not wear-resistant. When adding copper to the titanium alloy, copper oxide would be formed which is a gas volatilization. It will effectively consume the oxygen during the friction process especially during high temperature friction. As the temperature increases, the oxygen in the copper oxide volatilizes in the form of a gas, so that the remaining copper element will take oxygen from the titanium oxide to form copper oxide, and then the amount of the hard particle titanium oxide is reduced. In the process of increasing temperature, the tendency of titanium oxide to convert to copper oxide always exists, while the amount of hard particles becomes less and less, thus the friction coefficient is also getting lower. As the copper content increases, the amount of titanium oxide deceases, thereby improving the wear resistance.

The grinding ratios of the Ti-xCu (x = 2.5, 7, 14) alloy are shown in Figure 5b. Compared with Ti-2.5Cu (3.5%), the grinding ratio of Ti-7Cu (4.2%) and Ti-14Cu (6.9%) increases by 20% and 97%, respectively, which proves the addition of the copper markedly improves the grindability of the Ti-Cu alloys. The grinding ratio here refers to the mass change of the alloy before and after the wear test. It can be expressed by the following formula, Equation (1):(1)$$\mathrm{Grinding}\;\mathrm{ratio}=\frac{m_{a}-m_{b}}{M_{a}-M_{b}}$$
where $m_{a}$ is the mass of the grinding wheel to be tested before the friction and wear test, and $m_{b}$ refers to the mass of the grinding wheel to be tested after the friction and wear test. $M_{a}$ is the mass of the material to be tested before the friction and wear test, $M_{b}$ is the mass of the material to be tested after the friction and wear test. It can be seen from the test results that Ti_2_Cu precipitation is easier to form in Ti-Cu alloys with higher copper concentration (7 or 14 wt.%) along the grain boundary. This phenomenon may be caused by the better enrichment of copper in the grain boundary due to the lamellar structure of the Ti_2_Cu phase. The Ti_2_Cu precipitate enriched at the grain boundaries increases the number of grain boundaries, increasing the strength of the material and thus reducing the ductility. Therefore, the Ti-14Cu alloy had better grindability compared to Ti-2.5Cu and Ti-7Cu alloys.

### 3.3. Effect of Ti_2_Cu on Mechanical Properties and Tribological Properties

Figure 6 shows the structures of the Ti-Cu alloys.

Based on the XRD measurement, the intermediate phase that forms the eutectoid structure with α-Ti was identified as FCC Ti_2_Cu, and no primary β phase was formed. With the increase of Cu element in Ti-Cu alloys, the morphology of Ti_2_Cu phase changes from granular through mixing granular and short strip and then to acicular structure. The size of the Ti_2_Cu phase was measured using Image Pro Plus 6.0 and shown in Table 3.

It is worth noting that the Ti_2_Cu phase of Ti-14Cu reaches 18.874 µm, which increases by 54.413% and 91.639%, respectively, compared with Ti-7Cu and Ti-2.5Cu.

It also can be found that the Ti_2_Cu precipitates of the Ti-7Cu alloy distribute in grain and along grain boundaries, while in the Ti-14Cu alloy, the precipitates only distribute along the grain boundaries, which is all existing in the form of layer structure. Under the action of shear force and friction conditions, Ti_2_Cu phase with rigidity can improve the plastic shear resistance of Ti-Cu alloys, so as to improve its wear resistance. Ti-14Cu shows the best wear resistance could be caused by the existence of layered Ti_2_Cu.

In order to further determine the distribution of Cu elements, EDS was used for determination and the results are shown in Table 4.

It is found that the distribution in Ti-2.5Cu, Ti-7Cu and Ti-14Cu alloys was similar in grain boundary. The Cu content in the grain boundary is higher than that in intracrystalline. It is known that copper as a *β*-stable element, can promote the precipitation of Ti_2_Cu hard phase in titanium alloy. The higher copper content the alloy has, the greater fraction of the Ti_2_Cu hard phase precipitation in the alloy. Especially the needle-like Ti_2_Cu structure increases the effect of Ti_2_Cu particles, through increasing dislocation density, thus effectively preventing grain boundary migration, further improving strength and hardness. The interwoven and alternating lamellar Ti_2_Cu structure, which greatly increases the enrichment of Ti_2_Cu phase at the grain boundary, making the number of Ti_2_Cu phase at the grain boundary much larger than that inside the grain. Due to the higher copper content in the Ti_2_Cu phase, serious segregation phenomenon can be found in the Ti_2_Cu phase. With the increase of Ti_2_Cu phase segregation, the wear depth generated in the friction process becomes shallower, all as to withstand greater loads under normal friction-resistant pressure.

A schematic diagram of tensile specimens of a Ti-Cu alloy is show in Figure 7a, and cross sectional TEM images and the Cu distribution of the Ti-14Cu and Ti-2.5Cu alloys are shown in Figure 7b,c, respectively.

As can be seen from the Cu element distribution diagram in purple color, the hypereutectoid Ti-14Cu has a darker color and long strip structure, comparing with the granular structure of subeutectoid Ti-2.5Cu. It can be concluded that Ti_2_Cu phase precipitation increases with the increase of copper content. The mechanical behavior of Ti-Cu alloys depends directly on their microstructural characteristics, especially the nature of stabilized phases of α-Ti and Ti_2_Cu precipitates, which depends on the Cu content and processing routes applied.

Two assumptions have been proposed to explain the strengthening mechanism of the Ti-Cu alloys. One is related to the formation of a larger volume fraction of martensitic structure under high cooling rate. It is known that the increase of the content for β stabilizing element causes hexagonal martensite (*α’*) to transform into orthorhombic martensite (*α”*) [24,25], which ordinarily occurs in the alloy quenched from high-temperature with different processing routes. The other is related to precipitates strengthened by the Ti_2_Cu precipitates with various volume fraction, morphologies and distributions [25,26].

In this work, the main results show that the strength of the material is related to the precipitated strengthening effects caused by Ti_2_Cu precipitates having different volume fractions and morphologies. Higher Cu concentrations contribute to segregation phenomenon of Ti_2_Cu. A modified Orowan mechanism for Ti-Cu alloys to evaluate Ti_2_Cu precipitate effects was proposed [27,28] as the formula, Equation (2):(2)$$\Delta \tau =\frac{Gb}{2\pi \sqrt{1-\nu }\left(0.825\sqrt{\frac{d_{t}t_{t}}{f}}-0.393d_{t}-0.886t_{t}\right)}\times \mathrm{In}\frac{0.886\sqrt{d_{t}t_{t}}}{b}.$$
where Δ*τ* is the increment in yield strength for the alloy due to precipitate hardening, *b* is magnitude of Burgers-vector, *d_t_* is uniform diameter of precipitates, *t_t_* is thickness of precipitates, *G* is shear modulus, *f* is volume fraction of precipitates, *ν* is the Poisson ratio. Considering that copper element will increase the precipitation amount of Ti_2_Cu phase, the addition of Cu in this study will lead to an increase in the volume fraction of Ti_2_Cu precipitates, thus leading to an increase in the intensity. Furthermore, the increased strength shows a linear relationship with Cu concentration due to the difference in grain size and defect density which can also affect the material’s strength. Overall, the predictions are in good agreement with those obtained in the present work.

The wear scratch morphology of Ti-2.5Cu, Ti-7Cu and Ti-14Cu are exhibited in Figure 8.

It was found that the titanium alloy of Ti-2.5Cu had deep wear marks and obvious adhesive wear. The surface film would break and present a new metal surface under the action of normal force and tangential force during the friction process due to the low content of Ti_2_Cu. It is known that under the action of load, there will be a certain amount of wear. When the amount of wear accumulating to a certain extent, it would accumulate in the scratch, where larger wear particles will impact the wear surface and cracks occur, as shown in Figure 8b. With the increase of Cu content, the wear depth becomes shallower due to the increase of Ti_2_Cu phase. Ti_2_Cu can resist the normal compressive stress of the friction pair under the condition of friction, and bear a larger load accompanied by a little adhesive wear to resist the wear of the hard particles. It has been proved that the gradual accumulation of the Ti_2_Cu phase results in a lower degree of wear reduction.

## 4. Conclusions

In this work, the effect of different copper contents (2.5 wt.%, 7 wt.% and 14 wt.%) on the mechanical properties and tribological properties of titanium alloys was studied. The main conclusion are as follows:

The tensile strength and hardness of Ti-Cu alloys is increased with the increase of copper content. Compared with the tensile strength of Ti-2.5Cu alloy, YS of Ti-14Cu alloy is increased by 30% and UTS is increased by 25%. Ti-7Cu has 22% higher hardness than Ti-2.5Cu alloy, and Ti-14Cu has 43% higher hardness than Ti-2.5Cu alloy.

Compared with Ti-2.5Cu alloy, the grinding comparison of Ti-14Cu alloy is increased by 97%. It is proved that the addition of copper improves the wear resistance of Ti-Cu alloys. The elongation of Ti-Cu alloys decreased with the increase of Ti_2_Cu precipitation amount with the Ti-14Cu alloy exhibiting the lowest elongation. The fracture morphologies of Ti-Cu alloys are intergranular fracture and cleavage fracture, and the dimples appeared in 2.5% and 7% Ti-Cu alloys.

Copper as a β-stable element in titanium alloys can effectively promote the precipitation of the Ti_2_Cu hard phase. The higher the copper content, the greater the precipitation of the Ti_2_Cu hard phase. EDS results show that Cu concentration at the grain boundary is much higher than that inside the grain. The copper-rich zone at the grain boundary will greatly increase the segregation of Ti_2_Cu phase at the grain boundary. Ti_2_Cu phase with copper-rich region and needle-like structure improves the properties of titanium alloys to some extent. This work proves that adding proper amount of copper to titanium alloys can improve their mechanical and frictional properties, which might meet their application requirement in different fields such as biomedical and aerospace.

## Figures and Tables

**Figure 1 materials-13-03411-f001:**
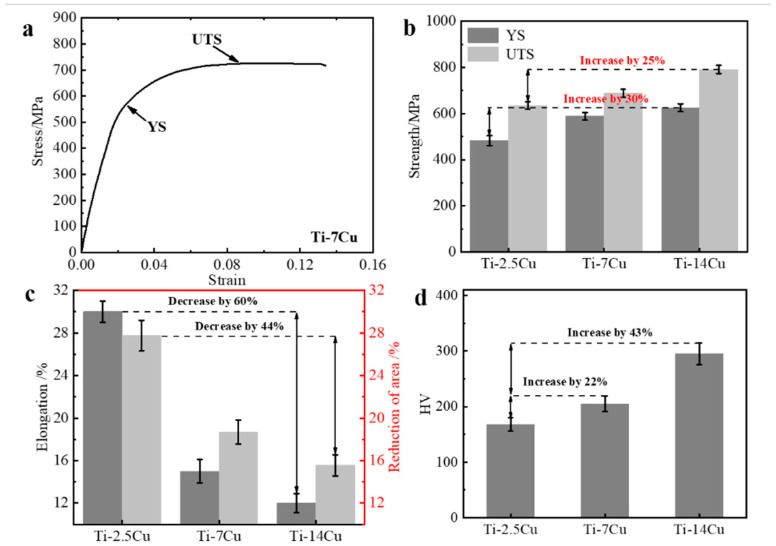
Tensile curves and mechanical properties of Ti-Cu alloys at room temperature. True stress–strain curves of Ti-7Cu alloy (**a**), and strength (**b**), elongation (**c**), hardness (**d**) of Ti-Cu alloys.

**Figure 2 materials-13-03411-f002:**
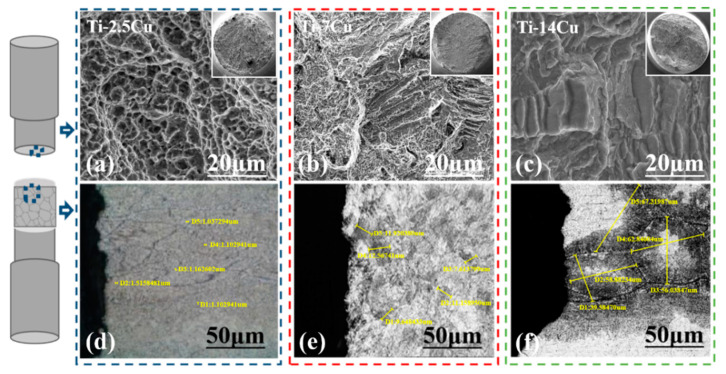
Fracture morphology of Ti-Cu alloys Ti-2.5Cu (**a**), Ti-7Cu (**b**) and Ti-14Cu (**c**), and section morphology of Ti-2.5Cu alloy (**d**), Ti-7Cu alloy (**e**), Ti-14Cu alloy (**f**).

**Figure 3 materials-13-03411-f003:**
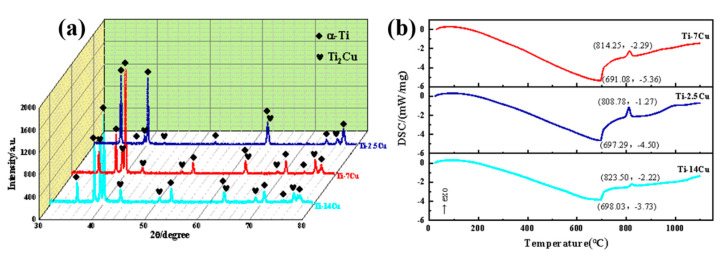
(**a**) XRD patterns of titanium alloys with different Cu contents; (**b**) differential scanning calorimetry (DSC) heating curves of titanium alloys with different Cu contents.

**Figure 4 materials-13-03411-f004:**
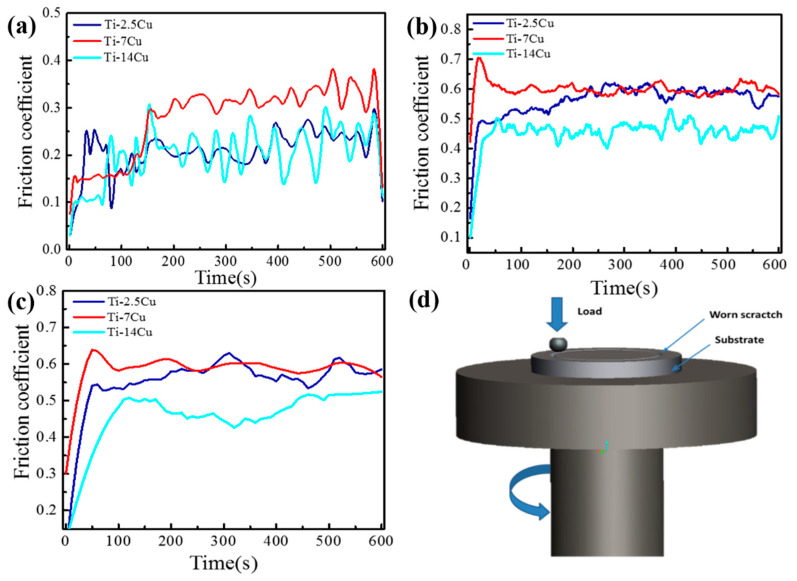
Friction coefficient of Ti-Cu alloys at (**a**) room temperature, (**b**) 400 °C, (**c**) 600 °C and (**d**) a pin-disk wear machine (MMQ)-reciprocating friction and wear test diagram.

**Figure 5 materials-13-03411-f005:**
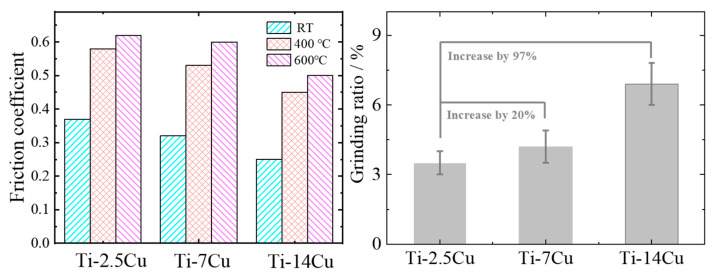
(**a**) Average friction coefficient of the three Ti-Cu alloys at different temperatures; (**b**) grinding ratio of Ti-Cu alloys.

**Figure 6 materials-13-03411-f006:**
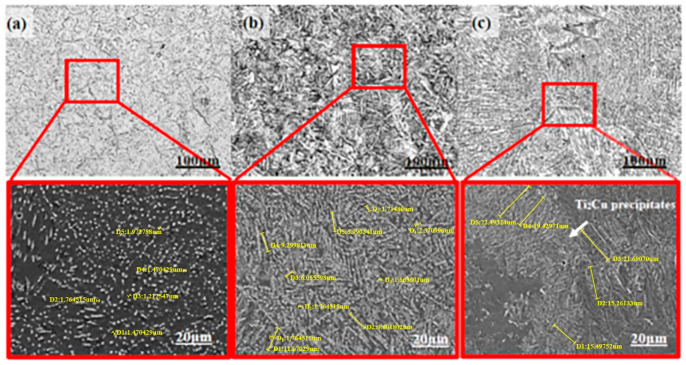
Microstructure of Ti-Cu alloys; (**a**) Ti-2.5Cu, (**b**) Ti-7Cu, (**c**) Ti-14Cu. Red frame: SEM images of grain boundary for SEM images.

**Figure 7 materials-13-03411-f007:**
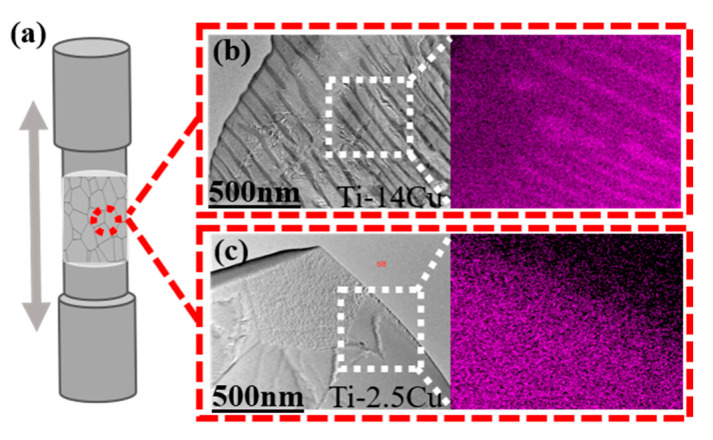
(**a**) Schematic diagram of tensile specimens of a Ti-Cu alloy; (**b**) cross sectional TEM image of Table 4. Cu and distribution diagram of the corresponding Cu element; (**c**) cross sectional TEM image of Ti-2.5Cu and distribution diagram of the corresponding Cu element.

**Figure 8 materials-13-03411-f008:**
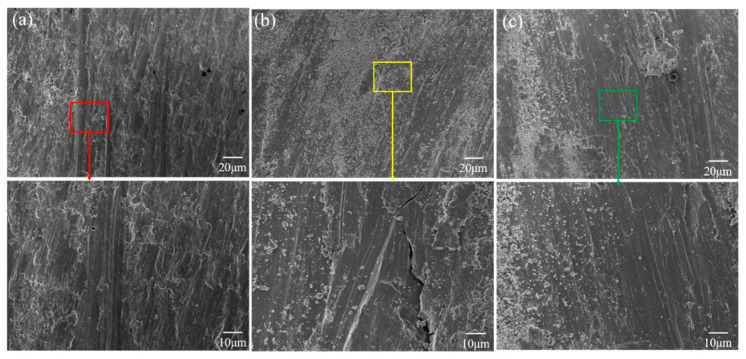
SEM images showing the microstructure and morphology of the worn scratch (**a**) Ti-2.5Cu, (**b**) Ti-7Cu and (**c**) Ti-14Cu.

**Table 1 materials-13-03411-t001:** Compositions of the Ti-Cu alloys.

Ti-Cu Alloy	Cu wt.%	Al wt.%	Si wt.%	O wt.%	N wt.%	C wt.%	Ti wt.%
Ti-2.5Cu	2.5	0.3	0.7	0.05	0.009	0.02	Bal
Ti-7Cu	7.0	0.3	0.7	0.05	0.009	0.02	Bal
Ti-14Cu	14.0	0.3	0.7	0.05	0.009	0.02	Bal

**Table 2 materials-13-03411-t002:** The calculated grain size of Ti-Cu alloys (D: µm).

Ti-Cu Alloy	D1	D2	D3	D4	D5	Average
Ti-2.5Cu	1.103	1.516	1.163	1.103	1.104	1.198
Ti-7Cu	9.648	11.159	7.614	12.567	11.920	10.582
Ti-14Cu	39.585	58.882	56.038	62.846	67.320	56.934

**Table 3 materials-13-03411-t003:** Size of the Ti_2_Cu phase of Ti-Cu alloys (D: µm).

Ti-Cu Alloy	D1	D2	D3	D4	D5	Average
Ti-2.5Cu (granular)	1.470	1.765	1.213	1.470	1.973	1.578
Ti-7Cu (granular) Ti-7Cu (short strip)	1.765 11.070	1.765 9.602	1.664 6.056	2.371 9.300	1.715 6.990	1.856 8.604
Ti-14Cu (acicular)	15.498	15.261	21.690	19.430	22.493	18.874

**Table 4 materials-13-03411-t004:** Concentration of Cu on grain boundaries and within grains.

Concentration of Cu (wt.%)	Grain Boundaries					Intracrystalline
	1	2	3	4	Average	
Ti-2.5Cu	4.32	3.18	3.76	3.38	3.66	2.21
Ti-7Cu	8.58	7.91	9.44	9.17	8.78	7.3
Ti-14Cu	22.17	18.23	18.34	23.02	20.44	12.7

1–4 means different EDS points on grain boundaries.

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
