# Peer review of "The Effect of Copper Content on the Mechanical and Tribological Properties of Hypo-, Hyper- and Eutectoid Ti-Cu Alloys"

_materials, 2020, doi:10.3390/ma13153411_

Round 1

Reviewer 1 Report

The authors have analyzed the effect of Cu content on the properties of hypo, hyper, and eutectoid Ti-Cu alloys. The manuscript is poorly written and must be considered for further processing after major revision. It is recommended to modify some points, as shown below;

(I) The abstract is very loose and should be modified to follow the reader's interest. Also, some grammatical errors have been found.

(ii) In the introduction section, reference should be provided for hypo, hyper, and eutectoid Ti-Cu alloys.

(iii) In the experimental part, the manufacturer of the initial materials was not seen. It should be provided. What was the content of oxygen, nitrogen, and carbon?

(iv) results and discussion section: first paragraph should not be started from references. The flow of the presentation should be changed. Such as; your results and reference in support ...

(v)Figure 1: what was stress /strain .. true or engineering...? Mention

(vi) The organization of this manuscript is cumbersome and therefore, need to revise carefully.

(vii) The review of current progress and analysis of drawbacks behind them are not well prepared.

(viii) Is there any reason at room temperature friction coefficient is very noisy but at higher temperatures it is comparatively smooth.?

(ix) TEM results were seen in Fig. 6 in the manuscript but there is no information in the experimental part.

Reviewer 2 Report

As stated in the Introduction, as result of literature review, structural state is very important for the properties, and for example significant difference could be expected between cast or thermo-mechanically treated samples. Therefore, is surprising that the authors give so little information about the fabrication route of the samples and the subsequent treatments.

Microstructural investigations should be considerably deepened with more qualitative and quantitative information, for example size of grains and/or precipitations, different morphologies, comparison between longitudinal and cross sections etc.

Phase identifications should be made by XRD to confirm references to precipitates, while relation to eutectoid may be documented with some DSC.

Otherwise there is little added value, if only chemical composition is related with the properties. Copper distributions by EDS seem promising, if correlated why other microstructural elements, and may provide some interesting conclusions regarding changes of properties.

Reviewer 3 Report

The manuscript is devoted to the study of the microstructure, mechanical and tribological properties of alloys of the titanium-copper system depending on the concentration of copper in the alloy.

However, further consideration of the manuscript is possible after correcting a number of comments.

1) Line 37. “Copper is a β-stabilizing element for titanium alloys”. However, according to the Ti-Cu phase diagram, this stabilization is characteristic only for a certain temperature range, it is possible that it is necessary to describe this moment in more detail.

2) Line 44. “the addition of Cu to Ti alloys can lower the melting point of the alloy”. According to the Ti – Cu phase diagram, a significant decrease in the melting point of the alloy is characteristic of large copper concentrations. It may be worth bringing the temperature data on the liquidus line to 2.5; 7 and 14 wt. % copper.

3) Lines 54, 55, 57, 282. “Alexandra [12] and Kikuchi [13]”; “Ohkubo et al. [14]"; “Kikuchi [14, 34] and Hayama et al. [17]. " Links do not correspond to the list of references. Please check.

Here is what you have in your bibliography:

  1. Souza, S.A., Afonsn, C.R. Ferrandinni, P.L. Effect of cooling rate on Ti-Cu eutectoid alloy microstructure…
  2. Andrade PN, Coelho AA, Afonso CRM, Contieri RJ, Robert MH, Caram R. Effects of composition on ...
  3. Li, J., Yu, N.N., Jiang, H.W., Leng, J.F., Geng, H.R. Effects of annealing temperature on dealloying of Ti - Cu ...
  4. Jiang, S., Peng, R.L., Jia, N., Zhao, X., Zuo, L., Microstructural and textural evolutions in multilayered Ti / Cu ...

4) Materials and preparation. It is necessary to describe in more detail the process of smelting samples (furnace grade, temperature, etc.), as well as the forging process (temperature and load, etc.). Was there only quenching in water and no more heat treatment?

5) Materials and preparation. The brand of the optical microscope is indicated, but the manuscript does not contain the results of a study with an optical microscope. Is it worth bringing a brand? But there are results from a transmission microscope, but there is no brand, you must specify the brand TEM.

6) Is there any XRD phase analysis data?

7) Line 129. Figure 2 (d), (e), (f). No caption, add it.

8) Figure 2 (d), (e), (f). The grain size is not clear, it is better to add a description of the microstructure in the text. Also add etching techniques (reagents, etc.) to materials and methods.

9) Figure 2 (b). The fracture is a mixed nature of destruction, correct it in the text.

10) Line 115 and 116. “Ti-5Cu to Ti-15Cu varies from 522 to 592 MPa”. Is there a typo in the brand of the alloys or is it literary data? Then you have to refer to the literature.

11) Was the percentage of Ti2Cu over the thin section area calculated?

12) Figure 1 (b), (c), (d) and Figure 4 (a). Strength increases, hardness increases, elongation decreases, but the friction coefficient for 2.5 Cu and 14 Cu at RT (room temperature) is the same. Data inconsistency, explain or check the data again.

13) Figure 7 (b). There is a crack in the figure, explain this in the text.

Round 2

Reviewer 1 Report

The present version of the manuscript is significantly improved from its original state. It can be accepted in present version.

Author Response

Thank you very much for your comments.

Reviewer 3 Report

The manuscript is devoted to the study of the microstructure, mechanical and tribological properties of alloys of the titanium-copper system, depending on the concentration of copper in the alloy.

The authors made corrections according to the comments.

However, there are a few things left that need to be clarified:

1) Did you use ready-made Ti – Cu alloys?

2) Figure 3. Does the DSC curve represent heating or cooling?

3) Table 2. How do you explain the large difference in grain sizes of alloy samples? Is there literature data on grain sizes of alloys of the titanium-copper system?
